# Uncovering Phytotoxic Compounds Produced by *Colletotrichum* spp. Involved in Legume Diseases Using an OSMAC–Metabolomics Approach

**DOI:** 10.3390/jof9060610

**Published:** 2023-05-25

**Authors:** Pierluigi Reveglia, Francisco J. Agudo-Jurado, Eleonora Barilli, Marco Masi, Antonio Evidente, Diego Rubiales

**Affiliations:** 1Institute for Sustainable Agriculture, CSIC, 14004 Cordoba, Spain; preveglia@ias.csic.es (P.R.); q92agjuf@uco.es (F.J.A.-J.); ebarilli@ias.csic.es (E.B.); 2Department of Chemical Sciences, University of Naples Federico II (UNINA), 80126 Naples, Italyevidente@unina.it (A.E.); 3Institute of Sciences of Food Production, National Research Council (ISPA-CNR), 70125 Bari, Italy

**Keywords:** *Colletotrichum* spp., fungal metabolites, anthracnose, legumes, phytotoxins, metabolomics, chemotaxonomy

## Abstract

Different fungal species belonging to the *Colletotrichum* genus cause anthracnose disease in a range of major crops, resulting in huge economic losses worldwide. Typical symptoms include dark, sunken lesions on leaves, stems, or fruits. *Colletotrichum* spp. have synthesized, in vitro, a number of biologically active and structurally unusual metabolites that are involved in their host’s infection process. In this study, we applied a one strain many compounds (OSMAC) approach, integrated with targeted and non-targeted metabolomics profiling, to shed light on the secondary phytotoxic metabolite panels produced by pathogenic isolates of *Colletotrichum truncatum* and *Colletotrichum trifolii*. The phytotoxicity of the fungal crude extracts was also assessed on their primary hosts and related legumes, and the results correlated with the metabolite profile that arose from the different cultural conditions. To the best of our knowledge, this is the first time that the OSMAC strategy integrated with metabolomics approaches has been applied to *Colletotrichum* species involved in legume diseases.

## 1. Introduction

Legumes are crops that belong to the family Fabaceae, characterized by their high protein content, being important for human food and animal feed, and by their ability to fix nitrogen from the atmosphere in symbiosis with nitrogen-fixing soil bacteria, which is key to the sustainability of cropping systems [1]. As with any crop, legumes are vulnerable to various biotic stresses, including fungal pathogens. These pathogens can cause devastating losses in crop yields, resulting in significant economic and environmental impacts [2]. Anthracnose, caused by pathogen *Colletotrichum* spp., is one of the most economically significant diseases affecting legumes. *Colletotrichum* spp. can survive for several years on plant debris that remains in the field after harvest [3]. Environmental high humidity rates and temperatures up to 20 °C are required by these pathogens to infect the host plant [4]. The initial symptoms on leaves are small yellow spots that enlarge into brown-colored lesions with a distinct dark margin. This might result in premature leaf drop. In the stem, the first lesions appear in its base from where they progress upwards [3]. Large stem lesions can cause wilting with subsequent plant death. In susceptible genotypes, more than 20% of the harvested seeds could show necrotic lesions, affecting their quality and market sale [5,6]. Among others, *Colletotrichum truncatum* and *Colletotrichum trifolii* are some of the most common fungal pathogens causing anthracnose in legumes. *C. truncatum* is a widespread pathogen that affects a range of legumes, including soybean, pea, and lentil [7,8,9], while *C. trifolii* primarily affects fodder legumes such as clover and barrel medic [10,11]. It has been observed that the host specificity of *Colletotrichum* isolates varies with the plant species from which they are obtained, and isolates from a single host also exhibit pathogenic variation [12].

Fungal pathogens are known to produce a range of phytotoxic metabolites that could be involved in the development of disease symptoms in plants, or they could be virulence factors contributing to fungal pathogenicity [13,14,15]. These metabolites belong to different classes, including cyclohexanones, macrolides, polyketides, terpenes alkaloids, and peptides. Understanding the molecular mechanisms behind the production of these metabolites can lead to developing new and more effective control strategies for fungal diseases [16,17,18]. The lifestyles of the various *Colletotrichum* species vary from necrotrophic to hemibiotrophic [19], producing a significant number of secondary metabolites, some directly contributing to their pathogenicity [4]. Over the decades, 189 secondary metabolites isolated from *Colletotrichum* spp. have been chemically and biologically characterized [20]. Nevertheless, only a few data are available on phytotoxic metabolites produced by *C. truncatum* [21,22]. While it is reported that *C. trifolii* can produce phytotoxic exopolysaccharides, no data are available on the isolation and characterization of phytotoxic low-molecular-weight metabolites. 

The production of phytotoxic secondary metabolites is affected by environmental factors such as nutrient availability, temperature, and pH [23]. When altering these factors, it is fundamental to fully explore the structural diversity of secondary metabolites of fungal pathogens in order to induce the production of specific biomarkers that can also be useful for chemotaxonomy application [24,25]. In recent years, the one strain many compounds (OSMAC) approach has emerged as a powerful tool for exploring the chemical diversity of fungal secondary metabolites. This approach involves manipulating growth conditions, such as pH, temperature, and media composition, to elicit diverse secondary metabolites from a single fungal isolate, expanding its chemical repertoire [26]. OSMAC has also been applied in integration with metabolomics [27]. Indeed, untargeted and targeted metabolomics have emerged as essential tools for studying the chemical diversity of fungal metabolites [28]. Both approaches have successfully identified phytotoxic metabolites produced by fungal pathogens, including *Fusarium*, *Alternaria*, *Penicillium*, and *Aspergillus* species [28,29]. 

Considering this background, the primary aim of this work was to investigate the secondary phytotoxic metabolites produced by three *Colletotrichum* pathogens of economic importance isolated from infected legume plants: (i) *C. truncatum* C428 isolated from lentil (*Lens culinaris*); (ii) *C. truncatum* C431 isolated from soybean (*Glycine max*); (iii) *C. trifolii* C436 isolated from red clover (*Trifolium pratense*). The OSMAC strategy was integrated with targeted and non-targeted metabolomics profiling. Moreover, to assess if and how the cultural media could affect the phytotoxicity of *Colletotrichum* spp. extracts, their activity was assessed on their primary host and related legumes. The secondary objective of this research was to assess the feasibility of host-specialized fungal strains from *C. truncatum* and *C. trifolii* that could be differentiated according to their metabolic profiles, paving the way to chemotaxonomy. To our knowledge, this is the first time the OSMAC strategy integrated with metabolomics approaches has been applied to *Colletotrichum* species involved in legume diseases. Moreover, it is the first time that phytotoxic metabolites have been dereplicated from an isolate of *C. trifolii*.

## 2. Materials and Methods

### 2.1. Fungal Strains, Plant Hosts, and Crossing Inoculations

Three previously well-characterized strains of *Colletotrichum* spp. kindly provided by Saskatoon Research and Development Centre (Canada) and maintained in the fungal collection belonging to the Institute for Sustainable Agriculture (IAS-CSIC, Córdoba, Spain) were used for the study (listed in Table 1). For the experiment, isolates were grown in Petri dishes containing potato dextrose agar (PDA) (Sigma Aldrich, Saint-Quentin Fallavier, France) under controlled conditions at 20 ± 2 °C under a 12 h photoperiod at 150 μmol m^−2^ s^−1^ photon flux density for 10 days, until sporulating mycelium was clearly visible. Then, a spore suspension for each fungal isolate was prepared by flooding the surface of 10-day-old cultures with sterile distilled water, gently scraping the colony with a glass rod and filtering the suspension through two layers of sterile cheesecloth. 

Disease responses were studied by performing cross-inoculation studies on a panel of legume species (listed in Table 2) that were grown under controlled conditions as follows: seeds were sown in pots (6 × 6 × 10 cm), filled with a potting mixture (sand/peat, 1:3 vol/vol), and kept in a growth chamber at 20 ± 2 °C and 65% relative humidity under a photoperiod at 14 h light/10 h dark with a light intensity of 200 μmol m^−2^ s^−1^ photon flux density supplied by high-output white fluorescent tubes, during a two-week period until the plants reached the 4–5-leaf stage. There were three independent replicates per fungal isolate and crop species, arranged in a completely randomized design. Each replicate consisted of 3 pots with 5 plants per pot. The experiments were repeated three times. 

The concentration of conidia from each spore suspension was determined with a hemocytometer and adjusted to 10^6^ spores/mL. Tween 20 (VWR) was added as a wetting agent (two drops per 500 mL suspension). The conidial suspensions were sprayed at the 4–5-leaf stage using a handheld sprayer at a rate of 1 mL per plant. A pot of each plant species was sprayed with sterile deionized water as the non-inoculated control. After inoculation, plants were covered with a polyethylene sheet during the first 24 h in darkness, and high humidity was ensured by ultrasonic humidifiers operating for 15 min every 2 h. Later on, the polyethylene cover was removed, and plants were maintained for 9 days in a growth chamber (under the conditions described above). Every 2 days, water was added to the trays to maintain high relative humidity (95–100%).

Anthracnose symptoms were rated for each fungal isolate on legume hosts ten days after inoculation. Disease response was assessed following a scale developed by Gossen [12] where lesions on the main stem of each plant were counted and grouped in categories as 1–10, 11–15, 16–20, 21–25, 26–30, and >30 lesions. Depth of penetration, position on the stem, and resulting degree of plant wilting were also integrated. As a result, numerical values were assigned to the levels within each descriptive category to permit statistical analysis. A final disease score was obtained by summing the values from each category [12].

### 2.2. Culture Medium and Growth Conditions for Fungal Organic Extract Production

For fungal organic extract production, each isolate was grown in four different culture media, as follows: (i) 6 Roux bottles (200 mL) containing each 100 mL of Richard’s medium [30]. Each bottle was inoculated with 3 mycelial plugs (of 5 mm^2^ each) of a 10-day-old mycelial plate of each isolate on PDA. The cultures were incubated at 24 °C, in constant stirring at 150 rpm, under a 12 h photoperiod at 150 μmol m^−2^ s^−1^ photon flux density for 21 days after which the mycelial mats were removed by filtration through four layers of filter paper, centrifuged and kept at −20 °C until further processing; (ii) 6 Roux bottles (200 mL) containing each 100 mL of potato dextrose broth (PDB) (BD Difco^®^, Crystal Lake, NJ, USA) medium. Each bottle was inoculated and kept as described in point (i); (iii) 6 flasks containing each 100 gr of normal rice substrate. Water was added to the flask (45%, vol/vol) and left for 24 h to be absorbed. Then, the material was sterilized at 121 °C for 30 min. The flask was then inoculated with 3 mycelial plugs (of 5 mm^2^ each) of a 10-day-old mycelial plate of each isolate on PDA. The cultures were incubated at 24 °C under a 12 h photoperiod at 150 μmol m^−2^ s^−1^ photon flux density for 30 days. Every 4 days, flasks were manually stirred to ensure fungal oxygenation. Then, the cultures were dried in a stove at 40 °C for 3 days and ground; (iv) 6 PDA Petri dishes were inoculated and incubated, as mentioned in point (iii), in static conditions for 10 days. 

### 2.3. Reagents and Materials

Solvents such as EtOAc, *n*-hexane MeOH, *i*-PrOH, CHCl_3_, and CH_2_Cl_2_ were purchased from Panreac AppliChem (Barcelona, Spain). Colletochlorin A, orcinol, and tyrosol were isolated from in vitro cultures of *Colletotrichum gloeosporioides* [31]; colletochlorins E and F, colletopyrone, higginsianin A, and higginsianin B were isolated from in vitro cultures of *C. higginsianum* [32,33]; (*R*)-mellein was isolated from in vitro cultures of *Neofusicoccum parvum* [34]; 6-hydroxymellein were isolated from in vitro cultures of *Phoma chenopodiicola* [35]; resorcinol was isolated from in vitro cultures of *Dothiorella vidmadera* [36]; and 4-hydroxybenzaldehyde and 2-(4-hydroxyphenyl) acetic acid were isolated from in vitro cultures of *Spencermartinsia viticola* [37]. The identity and purity of all standard metabolites were checked using NMR analysis at the Department of Chemical Sciences of the University of Naples Federico II. As an internal standard (IS) for the UHPLC-QTOF-HRMS analysis, (±)-naringenin analytical standard, purchased from Sigma-Aldrich, Milan, Italy, was used.

### 2.4. Extraction of Secondary Metabolites from Different Culture Media

The culture filtrates (180 mL) of the PDB and Richard’s media obtained from the three *Colletotrichum* isolates were exhaustively extracted with EtOAc (3 × 300 mL). The organic extracts of each strain were combined, dried (Na_2_SO_4_), and evaporated under reduced pressure yielding dark-red and dark-brown solid residues in the case of PDB and Richard’s medium, respectively. The solid rice cultures obtained from the three strains of *Colletotrichum* were air-dried as described in Section 2.2. The dried material (100 g) was minced using a laboratory mill and extracted with 100 mL of MeOH-H_2_O (1% NaCl) (1:1). The mixture was centrifuged for 1 h at 10,000 rpm. The pellet was extracted again with the same solvent mixture in the same conditions, and the two supernatants were then pooled, defatted by *n*-hexane (2 × 250 mL), and extracted with CH_2_Cl_2_ (3 × 250 mL). The CH_2_Cl_2_ organic extracts were combined, dried (Na_2_SO_4_), and evaporated under reduced pressure yielding dark-brown solid residues. The solid PDA cultures obtained from the three strains of *Colletotrichum* were lyophilized (Epsioln 2-10D, Christ, Osterod, Germany), then extracted with EtOAc (3 × 50 mL), and finally the organic extracts were filtered and evaporated under reduced pressure yielding dark-red solid residues. The media PDA, PDB, Richard, and rice substrates were extracted in the same specific conditions reported above, and used as blanks in the UHPLC-QTOF-HRMS. 

### 2.5. UHPLC-QTOF-HRMS Apparatus and Conditions

Before analysis, the 76 organic extracts were reconstituted in MeOH and filtered through a Millex syringe filter of 0.2 µm (Merck, Darmstadt, Germany), and the final concentration was 1 mg/mL. The internal standard (IS) was spiked in every standard solution at a 10 µg/mL concentration. Finally, 30 μL of each extract was taken to create 3 pooled QC samples.

The UHPLC-QTOF-HRMS analysis was carried out by the Metabolomics Platform of Agricultural Sciences, Food Science and Technology and Natural Resources (CEBAS)-CSIC, Murcia, Spain. Prior to mass spectrometric sample analysis, the chromatographic separation was realized using Acquity UPLC-I-class system (Waters Corporations, Milford, MA, USA). In total, 7 μL of the organic extract solutions was injected using a Sample Manager Fixed-Loop (SM-FL) (Waters Corporations, Milford, MA, USA). Chromatographic separation was performed using a Poroshell 120 EC-C18 Agilent column (100 × 3 mm, 2.7 μm, (Agilent Technologies, Waldbronn, Germany), operating at 30 °C and with a flow rate of 0.4 mL/min in an Acquity I-Class column oven system (Waters Corporations, USA). Compounds were separated using the following gradient conditions, using H_2_O + 0.1% formic acid (FA) (A) and MeCN + 0.1% FA (B): 0–10 min, 1–18% phase-B; 10–16 min, 18–38% phase-B; 16–22 min, 38–95% phase-B. Finally, the phase-B content was returned to the initial conditions (1%) for 1 min and the column was re-equilibrated for 5 min more. Software Compass HyStar (version 3.2 Bruker Daltonics, Bremen, Germany) was used for the operation of the UHPLC systems. The pooled quality controls (QCs) were used for metabolomic analysis quality. QCs, blanks, and (±)-naringenin solution (1 µg/mL) were injected three times during the batch test: beginning, middle, and end. A MaXis Impact QTOF mass spectrometer (Bruker Daltonics, Bremen, Germany) was utilized for QTOF-HRMS experiments. Ionization in the mass spectrometers was performed using an ESI source (Bruker Daltonics, Bremen, Germany), which operated under optimized conditions. The parameters for ESI source were set as follows: Ionization was performed in the negative mode at −4.0 kV. Dry gas temperature was set to 200 °C at a flow rate of 9.0 L/min. Nebulizer gas pressure was 2 bar. The ESI ion source was operated at −4.0 kV with a probe gas temperature of 450 °C at a flow rate of 4 L/min. The dry gas temperature was set to 300 °C at a flow rate of 9.0 L/min. The nebulizer gas pressure was 2 bar. A mass range of 50−1200 *m*/*z* was covered, and the full scan and MS2 data were recorded at a spectra rate of 2 Hz. Data-independent acquisition in the broadband collision-induced dissociation (bbCID) mode was chosen for MS/MS experiments. Fragmentation took place in a collision-induced dissociation cell using nitrogen. Spectral acquisition was performed at a collision energy (CE) of 20 eV. To calibrate the mass axis, a 10 mM sodium formate cluster solution in 1:1 isopropanol–water was introduced into the ESI source at the beginning of each UHPLC run using a divert valve to calibrate instrument mass calibration and re-calibrate individual raw data files. The content of identified Level A metabolites was quantified by the selected IS, and the relative peak areas (analyte area/IS area) were used for quantification. Software Compass Control (software version 3.4, Bruker Daltonics, Bremen, Germany) was used for the operation of the mass spectrometer, and for data acquisition and conversion of raw spectra files to centroid mzML files.

### 2.6. Metabolite Annotation and Identification

In order to fully exploit the differences in the metabolite profiles obtained in different media cultures of *Colletotrichum* spp., three bioinformatic tools were integrated: (1) MetaboAnalyst 5.0 [38], the web-based tool providing LC-HRMS spectra processing automated workflow integrated with comprehensive statistical data analysis and interpretation modules; (2) MS-DIAL (Version 5.10)/MS-FINDER (Version 3.5), the computational approach which helps to characterize the structure of the metabolites rapidly [39]; (3) MetFreg [40], a freely available web software to assist the annotation of the high-precision tandem mass spectra of metabolites by in silico fragmentation. 

This approach includes three main steps: Step 1: spectra processing and peak annotation with MetaboAnalyst 5.0. All the parameters are reported in Appendix A. Step 2: multivariate analysis of the global metabolites profile with MetaboAnalyst 5.0. Features with more than 50% of missing values in the samples were removed while missing values were estimated by a sample-wise k-nearest neighbors (KNN) algorithm. Datasets were normalized by the internal standard (±)-naringenin, and specific transformation and scaling conditions for the data sets are reported in Appendix A. The datasets were analyzed with PCA to explore potential patterns and heatmaps that could show clustering of the features and visualize the differences between groups. The top 20 (ANOVA *p* < 0.05) most significant features were selected for structural annotation. Moreover, PLS-DA was carried out to reveal the global profile changes and potential application for fungal classification according to metabolite composition. The VIP score cutoff value was 1.6. Cross-validation was carried out by LOOCV; R^2^ and Q^2^ values are reported in Appendix A. Step 3: structural annotation of the metabolites assisted by MS-DIAL/MS-FINDER and MetFrag. The metabolite ions were converted into structural information with MS-DIAL/MS-FINDER linked to MS/MS databases. The MS-DIAL parameters were MS1 tolerance of 0.01 Da; MS2 tolerance of 0.05 Da; and minimum peak height of 1000 (amplitude); for alignment, a QC sample was used as a reference file, and the retention time tolerance was set at 0.05 min. The MS/MS public databases used for peak identification were MSMS-Neg-MassBankEU, MSMS-Neg-GNPS, MSMS-Neg-MassBank, MSMS_Public_EXP_NEG_VS17, MSMS_Public_ExpBioInsilico_NEG_VS17, and MSMS-Neg-Vaniya-Fiehn_Natural_Products_Library, and the identification score cutoff value was selected as 8. Significant metabolites with monoisotopic mass error within ±5 mDa with no proper match in the selected databases were manually screened for mass spectral peak matching. The molecular formulas were searched on NpAtlas [41], Coconut [42], and Lotus [43] database to confirm whether they were previously produced by *Colletotrichum* spp. or by other microorganisms. SDF files were generated from the above-cited database and uploaded to MetFrag to assist in identifying the metabolites with in silico fragmentation. The complete list of identified/annotated metabolites and confidence levels is reported in Appendix A. The careful manual curation of all assigned peaks was carried out, and the metabolites were annotated according to confidence levels [44,45], also considering the available fungal taxonomical information.

### 2.7. Bioassays

A macroscopical evaluation of all the *Colletotrichum* spp. extracts was performed using a detached leaf method [46]. Legume species (listed in Table 2) were grown under controlled conditions, as described in Section 2.1. Four-stage leaves from each legume specie were excised and placed, adaxial side up, on 4% technical agar in Petri dishes. For each legume species, fungal extract, and concentration, cut leaves were arranged in a randomized design with three replicates per treatment, with each replicate having four leaves. Due to different solubilities, the four organic fungal extracts were dissolved in MeOH (5%) and then brought to the final concentration with distilled water (1 and 2 mg/mL). Two droplets of the test solutions (25 μL) were applied on the adaxial leaf side. Untreated leaves, droplets (25 μL) of distilled water, and droplets (25 μL) of MeOH (5%) were applied as negative controls. The solvent was evaporated in a laminar flow cabinet until dry. The presence of symptoms, consisting of leaf alteration by discoloration or necrosis, was periodically observed (from 1 to 14 days after the extract’s application). Total leaf area and leaf damaged area were measured in cm^2^ using the ImageJ (1.46 r) program (free license), and then the percentage of damage severity (%DS) was calculated. Additionally, the observed damage was classified as necrotic leaf area (N) or irregular discolored areas (DA) surrounded (or not) by a necrotic ring (NR). 

### 2.8. Bioassay Data Analysis

All experiments followed a completely randomized design. The significance of the differences in leaf damage between plant species, treatments, and concentrations was estimated by one- or two-way analysis of variance (ANOVA). Before performing ANOVA tests, the normality and equality of variances were checked using Shapiro–Wilk and Levene’s tests, respectively. Whenever the ANOVA test was statistically significant (*p* ≤ 0.05), a Fisher’s LSD multiple range test assessing the difference of the means between each treatment was performed. Null hypotheses were rejected when *p* ≤ 0.05. All statistical analyses were performed using the Statistix 9.0 package (Analytical Software, Tallahase, FL, USA). 

## 3. Results

### 3.1. Cross Inoculations

Cross inoculations performed with different *Colletotrichum* spp. showed that each isolate was more infectious to the legume species from which it was isolated, but that it also infected other legume species, although at a lower intensity. Isolate C428 of *C. truncatum*, collected from infected lentils, caused different severity indexes in the tested legume species (*p* < 0.01; Figure 1A), being most infectious to lentil (SI = 9.7), followed by white clover, pea, and faba bean (value ranging between 3.8 and 4.8), being even less infectious to subterranean clover, red clover, and soybean (SI < 2), and causing no symptoms in barrel medic (SI = 0) (Figure 1A). Isolate C431 of *C. truncatum*, collected from infected soybeans (*p* < 0.01; Figure 1B), was most infectious to soybean (SI = 9.3), followed by lentil, barrel medic, and white clovers (SI 5–4.2), and then by subterranean clover, red clover, pea, and faba bean (SI 3–1). Isolate C436 of *C. trifolii*, collected on red clover, was most infectious to red clover (SI = 9) (Figure 1C), followed by barrel medic and subterranean clover (SI 6.4–5.4), and then followed by faba bean, lentil, pea, and soybean (SI 3.5–3), and it was least infectious to white clover (SI = 1.6).

### 3.2. Targeted Analysis of Selected Phytotoxic Metabolites

Isolates C428 and C431 of *C. truncatum*, as well as isolate C436 of *C. trifolii* were grown in four different media: two solid substrates (PDA and rice) and two liquid media (PDB and Richard’s medium), as reported in detail in the Materials and Methods section. The production of 13 pure phytotoxic metabolites commonly produced by *Colletotrichum* species was investigated by LC-HRMS-ESI-QTOF in negative mode. The selected metabolites included the following: colletochlorins A, E, and F, colletopyrone, higginsianins A and B (*R*)-mellein, 6-hydroxymellein, orcinol, resorcinol, tyrosol, 4-hydroxybenzaldehyde, and 2-(4-hydroxyphenyl) acetic acid. Seventy-two extracts were analyzed in triplicate, and Compass software analyzed the quantitative data. The concentrations of the detected metabolites are reported in Table 3. 

Colletochlorins A and F, (*R*)-mellein, and tyrosol were not detected in any organic extract. Colletopyrone, higginsianin B, 6-hydroxymellein, and 4-hydroxybenzaldehyde were detected in all the organic extracts of all three *Colletotrichum* species; however, the amount varied according to cultural conditions. In detail, colletopyrone was detected in the highest concentration in the PDB cultures of C428 and C431 (8.13 μg/mL and 14.62 μg/mL, respectively), while, in C436, a quantifiable amount (0.18 μg/mL) was detected in rice extract. A quantifiable amount of higginsianin B (0.01 μg/mL) was detected only in the PDA culture of *C. truncatum* C428. 6-Hydroxymellein was quantified in the highest concentration in the rice cultures of both *C. truncatum* isolates, while it was below the LOQ in C436, regardless of the cultural conditions. 4-Hydroxybenzaldehyde was detected and quantified in all the organic extracts of the three pathogens. In particular, the highest amount was detected in Richard’s medium extracts. 

All three *Colletotrichum* species produced orcinol, but its yield depended on cultural conditions. Indeed, it was detected only in the PDB extract of *C. truncatum* C428. On the other hand, it was quantified (0.02 μg/mL) in the PDA extract of *C. truncatum* C431 and detected in the PDB extract. Moreover, it was detected in the organic extracts of the PDB and rice cultures of *C. trifolii* C436.

Resorcine was detected only in PDA (0.04 μg/mL) and in rice extracts of C428 and PDA and Richard’s medium extracts of C436, respectively. Colletochlorin E was detected only in Richard’s medium extracts of both *C. truncatum* isolates. In the extract of C428, colletochlorin E was also quantified at a rate of 0.01 μg/mL. Finally, higginsianin A was detected only in the PDA extract of *C. trifolii* C436.

### 3.3. Impact of Cultural Conditions on Secondary Metabolite Production by Colletotrichum Species

An untargeted metabolomics profiling was conducted to investigate the changes in secondary metabolite production due to in vitro cultural conditions. LC-HRMS-ESI-QTOF was used to compare the chemical profiles of the *Colletotrichum* species cultured. The LC–HRMS base peak chromatograms from the fungal cultures are shown in Appendix A. 

The LC-HRMS data pretreatment, followed by multivariate analysis to identify the up- and downregulated metabolites in each culture, was performed using MetaboAnalyst 5.0. Unsupervised data analysis by PCA was performed to visualize the data, look for trends and groupings, and to identify possible outliers. Heatmaps, which provide an intuitive data table visualization, were used to identify unusually high/low compounds in different cultural conditions. The analysis was carried out for each Colletotrichum species independently.

The results were then cross-confirmed by MS-DIAL, while MS-FINDER and MetFrag were used to identify putative fungal metabolites according to MS/MS data starting from those commonly produced by *Colletrotrichum* species, as fully described in Section 2.5 of the Materials and Methods. The complete list of identified/annotated metabolites, compound classes, and confidence levels is reported in Appendix A. The structures of validated/putatively annotated compounds Levels A, B(i), and B(ii) are reported in Appendix A. The results by pathogens are detailed and described in the sections below.

#### 3.3.1. *Colletotrichum truncatum* C428 (from Lentil)

The obtained peak intensity table containing the complete features detected from the raw spectra processing of isolate C428 was submitted to the statistical data analysis module of Metabonalyst. Data were normalized by the internal standard (I.S., naringenin), square root transformed, and scaled using Pareto scaling to reduce systematic bias and improve overall data consistency.

The PCA analysis highlighted the presence of four different groups mainly separated along the PC1 direction (Figure 2a). The extracts clustered according to cultural conditions with PDA and Richard’s medium extracts on the left quadrant of the score plot, while PDB and rice extracts are on the right quadrant (Figure 2a).

Among the compounds contributing to group discrimination, simple organic acids such as hydroxybutyric and succinic acids were produced in higher amounts in PDA and Richard’s medium, while dihydroxybutyric acid was upregulated in PDA (Figure 2b). Interestingly, the production of a compound with *m*/*z* 278.1035 and formula C_14_H_17_NO_5_ (putatively annotated as curvupallide, previously isolated from *Curvularia pallescens* [47]) was induced in both Richard’s medium and rice culture. The high-resolution mass and MS/MS spectra agreed with both curvupallides A and B. However, they differed only in the absolute configuration of OH at the C3; thus, it was not possible to discriminate between the two structures (Appendix A). 

When growing on rice, several secondary metabolites belonging to a different class of natural compounds were produced in higher amounts by *C. truncatum* isolate C428. Among them, phomolide B, a 10-membered macrolide previously isolated from *Phomopsis* sp., was putatively annotated [48]. Moreover, a compound with molecular formula C_25_H_29_N_5_O_7_ was classified as hybrid peptide-polyketide cyclic tri-depsipeptides, related to the class of the colletopeptides [49]. Finally, three compounds with molecular formulas C_23_H_31_NO_7_, C_23_H_29_NO_7_, and C_23_H_29_NO_8_ were classified as fusarins, mycotoxins produced by the *Fusarium* species [50]. The C_23_H_31_NO_7_ fusarin was also produced in high amounts in the PDB culture. 

#### 3.3.2. *Colletotrichum truncatum* C431 (from Soybean)

The obtained peak intensity table containing the complete features detected from the raw spectra processing of *C. truncatum* C431 was submitted to the statistical data analysis module of Metabonalyst. To reduce any systematic bias and to improve overall data consistency, data were normalized by the internal standard (I.S., naringenin), cube root transformed, and scaled using Pareto scaling.

In addition, for isolate C431, the PCA analysis highlighted the presence of four different groups. Nevertheless, differently from *C. truncatum* C428, PDA and Richard’s medium extracts were separated from the PDB and rice extracts along the PC1 direction, while PDA and Richard’s extracts could be discriminated along PC2. PDB and rice extracts could also be discriminated along PC2 (Figure 3a). Among the compounds contributing to group discrimination, hydroxybutyric and dihydroxybutyric acids were detected in higher amounts in PDA and Richard’s extracts (Figure 3b). Imidazoleacetic acid was upregulated in Richard’s medium and PDB (Figure 3b) Imidazoleacetic acid is an alkaloid, which was previously isolated from the mushroom *Coprinus atramantarius* [51]. Maculosin, a phytotoxic diketopiperazine that was previously isolated from pathogenic fungi and bacteria [52,53], was also upregulated in Richard’s and PDB medium. Berkchaetorubramine, a red pigment belonging to the family of azaphilones, was previously isolated from *Pleurostomophora* spp. [54], and detected in a higher amount in PDB extract. Moreover, another fusarin with molecular mass C_23_H_31_NO_8_ was detected in a high concentration in the PDB extract. Among the compounds upregulated in the rice culture, it was fascinating to denote the high concentration of phomolide B, together with another bioactive macrolide named Sch-725674, which was previously isolated from *Colletotrichum* spp. GDMU-1 isolated from the leaves of *Santalum album* [55] (Figure 3b).

#### 3.3.3. *Colletotrichum trifolii* C436 (from Red Clover)

The obtained peak intensity table containing the complete features detected from the raw spectra processing of *C. trifolii* isolate C436 was submitted to the statistical data analysis module of Metabonalyst, normalized by the internal standard (I.S., naringenin), logarithmic transformed, and autoscaled.

The PCA analysis of *C. trifolii* C436 highlighted that Richard’s medium extracts were separated along the PC1 from PDB and rice extracts, while it was possible to discriminate the PDA extracts along the PC2 (Figure 4a). PDA extracts were separated along PC1 from PDB and rice extracts; nevertheless, the latter extracts mostly overlapped in the score plot (Figure 4a). As shown in the heatmap (Figure 4b), among other compounds, PDB and rice substrate media shared the upregulation of two classes of secondary metabolites: colletopeptides and fusarins. Indeed, a colletopeptide with molecular formula C_26_H_29_N_3_O_6_ and fusarins with molecular formulas C_23_H_29_NO_7_, C_23_H_29_NO_8_, C_23_H_30_NO_7_, and C_23_H_31_NO_8_ were detected in higher concentrations. Moreover, another fusarin with molecular formula C_23_H_29_NO_7_ was upregulated only in the PDB medium (Figure 4b).

Hydroxylated organic acids also contribute to group discrimination. A trihydroxy octadecanoic acid was detected in high amounts in PDA, Richard’s medium, and rice extracts. In addition, hydroxybutyric acid was upregulated in PDA and Richard’s medium, while dihydroxybutyric acid was upregulated only in PDA. Finally, a tricarboxylic acid derivative with molecular formula C_12_H_20_O_7_ (Figure 4b) was detected in higher amounts in PDA and Richard’s medium and rice extracts.

### 3.4. Effects of Culture Media on the Phytotoxicity of Organic Extracts

The liquid culture filtrates of *Colletotrichum* spp. were exhaustively extracted. The symptoms occurring on treated leaves started to appear on the sixth day after inoculation, and the severity increased during the following days, being maximal at 12–13 days after inoculation. The organic extracts showed differential phytotoxic activities depending on the fungal isolate, the culture media, the tested concentration, and the host plant, as shown in Appendix A.

#### 3.4.1. *Colletotrichum truncatum* C428 (from Lentil)

In general, all the four tested fungal extracts from C428 showed significantly increased phytotoxicity (measured as DS%) compared with the controls (Figure 5a,b). In terms of legume species, lentil was a crop with significantly higher DS% values, regardless of the culture media employed for its growth or the concentration applied. Necrotic area as well as discolored tissues surrounded by a necrotic ring reached together nearly the 100% of the host leaves in response to fungal extracts from Richard, PDA, and PDB media (Figure 5c–e). Lower values were achieved with rice substrate medium and were dose-dependent (Figure 5f and Figure 6). Similarly, barrel medic also showed high DS% values, but only discolored areas surrounded by necrotic rings were observed. Soybean, faba bean, and pea were poorly affected by all of the culture filtrates, while all the clovers showed a discolored leaf area surrounded by a necrotic ring, with moderate to high DS% values (17 < DS% < 89.5) (Figure 6; Appendix A). Clovers were especially sensitive to Richard’s extracts, as well as to rice, with a dose-dependent effect. In general, C428 fungal exudate from Richard’s medium was the most phytotoxic. 

#### 3.4.2. *Colletotrichum truncatum* C431 (from Soybean)

Regardless of the culture media, the four tested fungal extracts from C431 showed phytotoxicity on soybean with necrosis and a discolored area surrounded by a necrotic ring, in spite of reduced DS values (<11%) (Figure 5i–l and Figure 6), and with effects that were not dose-dependent except for in Richard’s medium. Similar symptoms but higher DS% values were achieved by Richard’s medium extract in red clover, white clovers, and lentil (Appendix A), as well as by rice medium in red clover and barrel medic. In clovers and barrel medic, extended discolored areas are incited by PDA, with associated necrotic rings (in PDB-treated leaves), and with moderate to high DS% values. No symptoms, or only few necrotic spots, were developed with any of the C431 extracts on pea and faba bean. An overall difference in phytotoxic activity between culture media was not found. 

#### 3.4.3. *Colletotrichum trifolii* C436 (from Red Clover)

In clovers, all the fungal extracts from C436 caused impaired leaf tissues (discolored areas surrounded by a necrotic ring) compared with the controls (Figure 5). The presence of additional necrotic areas, as well as increasing fungal severity, depends on the cultural media and the concentration applied, with Richard’s and rice media being the most phytotoxic (DS% > 53 and 18 and DS% > 83 and 42, for Richard’s and rice medium at 1 and 2 mg/mL, respectively) (Figure 6). In terms of legume species, subterranean clover and barrel medic were the crops with significantly higher sensitivity in almost all media employed for their growth (Appendix A). In Richard’s medium, the diseased area reached up to 80% of the host leaves. Similarly, Richard’s medium was also phytotoxic to pea, faba bean, and soybean leaves, but with low to moderate DS% values (11 < DS% < 57). The symptoms in lentil were strongly influenced by the culture media, with the crop being highly susceptible to Richard’s medium at concentrations of 2 mg/mL, moderately susceptible to PDB and rice at 2 mg/mL, or poorly diseased following the application of PDA extracts at any concentration. 

### 3.5. Secondary Metabolite Profiles as a Tool for Chemotaxonomy of Colletotrichum Species

To explore the possibility of using the untargeted metabolomics profiles for chemotaxonomical purposes, partial least squares (PLS)-discriminant analysis (DA) was used to differentiate groups and identify the intrinsic variations in the data sets. The peak lists from each cultural media were reorganized, and the analysis was carried out using the *Colletotrichum* isolates/species as classes. As a result, the secondary metabolites specifically produced by *C. truncatum* C428 and C431 and *C. trifolii* C436 could discriminate between fungal species and isolates, regardless of the cultural media (Figure 7). A threshold of 1.6 was used for the VIP score. The normalization process of each data set and the Q2 and R2 values of each PLS-DA model, developed according to the cultural media, are reported in Appendix A. 

Figure 7a shows the score plot of the PLS-DA for the PDA extracts; only eight metabolites had a VIP score higher than 1.6, and half of them capable of discriminating between fungal species and isolates were organic acids and organic acid derivatives, including succinic acid, succinic anhydride, 3-nitropropionic acid, and an organic acid with an *m*/*z* of 115.0033 for which MS/MS data could be in agreement with both fumaric acid and maleic acid. The detection of 3-nitropropionic acid is particularly interesting because it is a potent antimicrobial agent produced by plants and fungi [56,57]. All of these acids were detected in higher amounts in *C. truncatum* C428. Colletopyrone also contributed to discriminating the fungal pathogens, being highly produced by *C. truncatum* C431 in this medium (Figure 7b).

When the PLS-DA analysis was carried out on PDB extracts, 15 metabolites having a VIP score > 1.6 contributed to the class discrimination (Figure 7c,d). Among them, two fusarins with formulas C_23_H_29_NO_6_ and C_23_H_29_NO_9_ and one colletopeptide with the formula C_31_H_33_N_3_O_7_S were upregulated in *C. trifolii* C436. Meanwhile, 3,4-dehydro-6-hydroxymellein, an isocoumarine previously isolated from *Ceratocystis minor* [58] and *Torrubiella tenuis* [59], together with a jasmonic acid derivative with a molecular formula of C_12_H_20_O_3_ were produced in a higher amount by *C. truncatum* C428. Several fungi have been reported to produce jasmonic acids to manipulate and/or hijack plant hormone defense signaling cascades for their own benefit [60,61]. Another isocoumarine, dereplicated as fusarentin 6,7-dimethyl ether and previously isolated from *Colletotrichum* spp. [62], together with methylsidowate, a sesquiterpenoid previously isolated from *Aspergillus* spp. [63], and berkchaetorubramine contributed to the discrimination being produced in higher amounts in this medium by *C. truncatum* C431. 

The score plot of PLS-DA analysis of the metabolic profiles of the rice extracts, reported in Figure 5e, shows three clear groups according to the *Colletotrichum* species and isolates. Eighteen compounds had a VIP score value higher than 1.6. Most of the discriminant metabolites were detected in higher concentrations in the extracts of *C. trifolii* C436. Interestingly, different from the PDB, the production of fusarentin 6,7-dimethyl ether is upregulated in *C. trifolii* C436 (Figure 7f). Moreover, three colletopeptides with formulas C_25_H_29_N_5_O_7_, C_28_H_42_N_4_O_5_, and C_29_H_45_N_5_O_5_, a fusarin with formula C_23_H_31_NO_8_, and colletol and colletodiol, two macrolides previously isolated from *Colletotrichum capsici* [64], were also detected in higher amounts by C436. Among the metabolites upregulated in *C. truncatum* C431 that also contributed to the class discrimination, two isocoumarines, alternariol, a mycotoxin previously isolated from *Alternaria* species [65] but also from *Colletotrichum* species [66], and 6-hydroxymellein were dereplicated (Figure 7f).

Finally, the secondary metabolite profile of Richard’s medium extracts allows for categorizing the three *Colletotrichum* according to isolates and species (Figure 7g). Fifteen metabolites showed a VIP score > 1.6 and contributed to the class discrimination (Figure 7h). Most discriminant metabolites were detected in higher concentrations in *C. truncatum* C431. Among them, berkchaetorubramine, TMC-205, an indole alkaloid previously isolated from unidentified fungal strain TC 1630 [67], pantothenic acid, also known as vitamin B5 and important for homeostasis and virulence of *Aspergillus fumigatus* [68], cytosporone T, an octaketides previously isolated from *Phomopsis* sp. IFB-ZS1-S4 [69], 4-hydroxybenzaldehyde and 4-chloro-3,5-dimethoxybenzaldehyde, a chlorinated orcinol derivative with antimicrobial activity and previously isolated from *Hericium erinaceum* [70], were also putatively annotated and upregulated in *C. truncatum* C431 (Figure 5h).

## 4. Discussion

Among the diseases affecting legumes, anthracnose, caused by fungal pathogens from the genus *Colletotrichum*, is one of the most economically significant [2]. Fungal isolates such as *C. truncatum* and *C. trifolii* affect different agronomically important grain and forage legumes [3], and their host specificity varies with the plant species from which they are obtained [71]. In the present work, we were studying the differential disease severity caused by three *Colletotrichum* isolates on a panel of different legumes, with a special focus on the phytotoxic metabolites produced by each pathogen, how these metabolites could be responsible for the exhibited host specificity, and how the in vitro growth conditions could affect their production. To the best of our knowledge, this is the first time that the OSMAC strategy, integrated with targeted and untargeted metabolomics approaches, has been applied to *Colletotrichum* species involved in legume diseases. Nevertheless, this strategy was previously applied to other fungal species [27]. Moreover, this is the first time that low-molecular-weight phytotoxins have been dereplicated from an isolate of *C. trifolii*.

It has been described that isolates of *C. truncatum* from lentil and from soybean produced distinct symptoms in their own hosts, as well as in different legume crops under controlled conditions. Our results on host specificity support the case for taxonomic separation of the pathogen *C. truncatum*, which is in agreement with previous studies [12,72]. Under controlled conditions, in the current study, the isolate from lentil was pathogenic to lentil, white clover, faba bean, and pea, while few or no symptoms were found in soybean and barrel medic, confirming previous findings [12]. In contrast, inoculation with the isolate from soybean produced visible lesions in soybean, and, to a lesser extent, also in lentils, barrel medic, and clovers. However, pea and faba bean were the hosts that developed the fewest symptoms of infection. This is also in line with previously developed studies on several hosts under both field and controlled conditions, where important differences in latent period and infection strategies between isolates from lentil and soybean were found [12,72]. Several reports indicate that anthracnose symptoms appear to be triggered by a change in host physiology, especially when plant tissues are under stress [73,74]. Indeed, many chemical and physical factors directly or indirectly contribute to metabolic pathway activation, including phytotoxic secondary metabolites produced by the fungus [26]. In this study, the metabolomics analysis performed on the fungal extracts, together with the chemometrics analysis, has pinpointed a total of 84 discriminant metabolites. In detail, 9 compounds were validated with pure standards (level A), 20 were putatively annotated (levels B(i) and B(ii)), and for 43 compounds it was possible to assign only the natural product class (level C(i) and C(ii)), while 12 remained unknown. As a general result, these secondary metabolites produced by *Colletotrichum* species belong to numerous natural products classes, including alkaloids, terpenoids, coumarins, chromones, xanthones, polyketides, quinones, peptides, phenols, and macrolides. These results agree with the capability of this fungal genus to produce a wide range of interesting bioactive compounds [4,20]. Furthermore, our data show that their production is heavily dependent on the selected cultural media.

Among specific *Colletotrichum* spp., all the fungal pathogens studied in this investigation produced colletopyrone, and higginsianin B, the identity of which was validated with pure standards. Moreover, their concentrations depend on the cultural conditions. Colletopyrone is a bioactive polyketide-derived compound that contains a fused six-membered and eight-membered ring system with a lactone and a ketone group. It is produced by various species of *Colletotrichum* and exhibits a range of biological activities, including phytotoxicity, cytotoxicity, and antifungal properties [20]. Higginsianin B, a diterpenoid α-pyrones, was previously isolated from *Colletotrichum higginsianum* and showed cytostatic activity against cancer cells [32]. Furthermore, 6-hydroxymellein was also detected in all the organic extracts, and its concentration depended on the cultural conditions. This compound belongs to the class of isocoumarins and is reported to have weak antimicrobial, cytotoxic, and phytotoxic activity [75]. Even though it was previously produced by several fungal species, this is the first time that has been reported in *Colletotrichum* species. Another interesting result that arose from the targeted analysis was that the production of colletochlorin E and higginsianin A seems to be species-specific and cultural medium-specific. Indeed, colletochlorin E was detected only in Richard’s medium extracts of *C. truncatum* isolates. This metabolite is tetrasubstituted pyran-2-one, previously isolated from *C. higginsianum*, and it showed only modest phytotoxic activity in *Sonchus arvensis* and tomato leaves [33]. On the other hand, higginsianin A was detected only in the PDA extracts of *C. trifolii* C436. This metabolite was isolated together with higginsianin B from *C. higginsianum* and showed cytostatic activity.

The chemometric analysis of the untargeted metabolomics profiles of the organic extracts showed that the selected cultural media—PDA, PDB, Richard’s medium, and rice—influence the production of specific secondary metabolites, and even though the upregulated or downregulated compounds depend on the *Colletotrichum* species, as reported in the results sections, some general patterns can be highlighted. Various small and/or hydroxylated organic acids, including succinic acid, fumaric acid/maleic acid, trihydroxy octadecanoic acid, hydroxybutyric acid, and dihydroxybutyric acid were upregulated in PDA and Richard’s media. It is important to underline that these compounds are primary metabolites, but they are usually released in the culture medium by the fungus. The optimization of their production and extraction process from fungi is an important challenge in the biotechnology field [76]. Nevertheless, pathogenic fungi can produce various small organic acids, including succinic, oxalic acid, or fumaric acid, which play important roles in fungal pathogenesis [77,78]. Moreover, the hydroxylated derivatives of fatty acids might have a role in fungal quorum sensing [79]. The production of colletopeptides and fusarins was upregulated when the *Colletotrichum* species were grown in PDB and rice media. This result was more evident for *C. trifolii* C436. Five colletopeptides were annotated at the level of confidence of C(ii). Several cyclic depsipeptides have been isolated from microorganisms, especially fungi in the genera *Aspergillus*, *Beauveria*, *Fusarium*, *Penicillium*, and *Colletotrichum* [80,81]. They show different biological activities, and they can play a role in the fungal defensive response. Seven fusarins were annotated at the level of confidence of C(ii). Fusarins are mycotoxins produced by *Fusarium* and other fungal species [82,83]. They are important virulence factors for *Fusarium* species, as they can suppress the host’s immune response and promote fungal colonization [82]. Moreover, their production could be influenced by various genetic and environmental factors [83,84]. Finally, the production of macrolides colletodiol, colletol in *C. trifolii* C436, and Sch-725674, and phomolide B in *C. truncatum* isolates was upregulated in rice culture. These results further support the capability of *C. truncatum* isolates to produce macrolides, as recently reported [22]. Fungi, including the *Colleotrichum* genus, are an important source of macrolides [20,85]. Various enzymes, including polyketide synthases and non-ribosomal peptide synthetases, mediate their biosynthesis. These compounds have diverse biological activities, including antimicrobial, antifungal, anticancer, and insecticidal properties, making them attractive targets for drug discovery and agricultural applications [86,87]. Regarding the latter, fungal diseases of crops of agronomic importance are usually chemically controlled by commercial fungicides from multiple chemical compound groups when no genetic resistance is available. However, pathogens have the ability to develop resistance to several chemical classes of compounds within a few years. This situation has decreased the efficacy of the major fungicides that are employed against crop pathogens, leading to the application of general integrated disease management strategies such as dose limitation, mixtures, and the continuous search for novel antimicrobial compounds [23]. Many plant pathogens, especially necrotrophic fungi such as *Colletotrichum* species, as we have seen here, are capable of producing a broad panel of natural substances representing an unexploited source of potential bio-fungicides with new molecular structures and modes of action against several crop diseases that should be tested in the near future. 

Overall, these results could be a useful starting point for more in-depth studies, while also integrating them with other omics techniques to shed light on the effects of cultural conditions on bioactive metabolite pathway activation in *Colletotrichum* species. The improvement in the culture conditions and a better knowledge of the regulation of secondary metabolism in *Colletotrichum* species are also essential in order to evaluate the impact on the phytotoxicity of extracts. Our results showed that their phytotoxicity varies depending on legume species, fungal isolates, cultural conditions, and the tested concentration. Furthermore, the high phytotoxicity of Richard’s medium extracts of *C. truncatum* C428 and *C. trifolii* C436 agrees with data previously reported [88,89,90]. These pathogens in this medium can produce some non-host-specific phytotoxins capable of inducing necrosis on all the plant species tested in our studies, even with different intensities. This variation in phytotoxicity of organic extracts on different plant species might be attributed to several factors, including differences in plant physiology and the chemical composition of the extract. Indeed, as highlighted by the metabolomics analysis, the chemical composition of organic extracts varied according to the cultural conditions, which affected their interaction with different legumes. This caused varied responses ranging from low effects to severe damage. Our results could help to select a specific cultural condition to investigate target phytotoxic metabolites involved in each pathosystem. For example, the severity of symptoms induced in lentil by *C. truncatum* C428 was higher for both PDA and Richard’s broth extracts, followed by PDB. Thus, the former extracts could be studied to investigate the role of small and/or hydroxylated organic acids in pathogenicity, while PDB extract could be investigated to shed light on the roles of colletopeptides and fusarins in disease development. On the contrary, for *C. truncatum* isolate C431, despite inducing high disease symptoms in a cross-inoculation assay, a low phytotoxicity of the organic extract was observed, regardless of the cultural conditions. These differences in symptom development, also observed for other legumes in a cross-inoculation assay, may be attributed to the pathogen’s ability to produce high-molecular-weight phytotoxins that remain unextracted under the used condition. These unextracted compounds could have a synergistic effect, exacerbating the observed symptoms [91]. Indeed, toxic exopolysaccharides and extracellular peptides have been isolated from other fungal species [92,93]. Hence, future investigations could explore the production of high-molecular-weight phytotoxins by *Colletotrichum* species to gain a better understanding of their role in pathogenicity. It is noteworthy that the highest phytotoxicity on all studied clover species for *C. trifolii* isolate C436 was observed with Richard’s medium extract. However, since the isolate was collected from red clover, which exhibited the highest severity index in the cross-inoculation assay, the PDA’s extract was the most specific, causing greater phytotoxicity in *Trifolium pratense* compared to other *Trifolium* species. Studying specific compounds could help us to fully understand the mechanisms behind host–pathogen interactions. Indeed, follow-up studies are underway to correlate the phytotoxic activity with specific metabolites. Nevertheless, the in vitro bioassay should be considered only as a starting point in investigating the role of secondary metabolites in symptom development. Only in planta studies can confirm their involvement in the pathogenicity or virulence, or if they have another role in the ecology of these fungi. 

Information on the chemotaxonomy of *Colletotrichum* spp. is minimal. Our results showed that secondary metabolites could help to classify the three *Colletotrichum* at both species and isolate levels. These results could assist species identification, enabling precise classification and enhancing our understanding of fungal diversity. Furthermore, they could support ecological studies by unraveling interactions between fungi and their host plants. In our research, C428 was isolated from lentils, while C431 was isolated from soybeans, and the observed metabolic differences may reflect the level of specificity in host selection. However, further in-depth studies are required to identify appropriate biomarkers for *Colletotrichum* spp. chemotaxonomy. This outcome is also essential for the selection of a proper phytotoxic metabolites pool for further in planta studies with the aims of (i) understanding the role of phytotoxins in the pathogenicity ad virulence of *C. truncatum* and *C. trifolii*, and (ii) selecting potential specific biomarkers for the early detection of anthracnose-assisted molecular methods.

At this stage, it is essential to highlight the importance of chemometric tools in metabolomics studies to fully explore the biological meaning behind the feature tables [94]. Indeed, applying a combination of univariate and multivariate analysis such as ANOVA, heatmap, PCA, and PLS-DA allows us to visualize hidden patterns in the data. Nevertheless, the PLS-DA algorithm should be regarded as just one building block in the steps used to develop a classification model, mainly when the study involves small sample sizes and large numbers of variables [95], such as the secondary metabolites produced by fungi. Furthermore, fungal metabolites’ vast chemical diversity and complexity contribute to the main bottleneck of untargeted metabolomics using LC-MS/MS: the unbiased structure assignment to metabolites of interest. In fact, despite outstanding progress made in the last decade in growing the number of metabolites in databases, many of the signals detected in metabolomics experiments cannot be directly assigned to specific metabolites because of the absence of their spectra in metabolomics databases [28,96]. This is particularly true for fungal metabolites due to the absence of wide and specific database spectra libraries. Modern dereplication strategies have been developed to tackle this problem, including machine learning, in silico fragmentation, or molecular networking [97,98,99,100]. Moreover, adding taxonomical information, or, in general, metadata from previous biological knowledge, and using a combination of bioinformatic tools, as was carried out in this investigation, could assist in obtaining a correct dereplication of the fungal metabolites [101,102]. Another significant limitation of this approach is that standard spectral library matching or in silico fragmentation cannot distinguish between potential stereo- and regioisomers, resulting in a level C annotation, or the fragmentation profile could agree with several classes, resulting in an unknown classification. This was also clear in our study, where more than 65% of the discriminant metabolites were level C or unknown. Dereplication strategies are fundamental to screen the crude extracts for the presence of known compounds and to identify compounds produced in a very low amount that could be undetected or lost during tedious purification steps. However, using orthogonal analytical methods, such as NMR, to validate metabolites, reach level A identification, and dereplicate unknown or characterize novel metabolites is unavoidable. Only by elucidating unknown metabolites, including assigning absolute or no relative configuration to chiral carbons, can we decipher complex biological systems.

## 5. Conclusions

Studying secondary metabolites, and in particular phytotoxic metabolites, is essential in order to gain information on the pathogenicity or virulence of a specific pathogen, and also in order to develop more sustainable control methods. Indeed, this research could be beneficial for investigations dealing with fungal chemotaxonomy [24], host–pathogen interactions and searching for potential fungal biomarkers [103,104,105], or, more generally, investigations dealing with genomics, transcriptomics, and proteomics in order to provide a more comprehensive understanding of fungal metabolism and physiology [106,107]. Finally, considering that *Colletotrichum* species are an important source of bioactive metabolites, as well as the continuous search for novel antimicrobial compounds and the increasing rate of fungal diseases worldwide, these data could also be used for multidisciplinary studies in life science and drug discovery.

## Figures and Tables

**Figure 1 jof-09-00610-f001:**
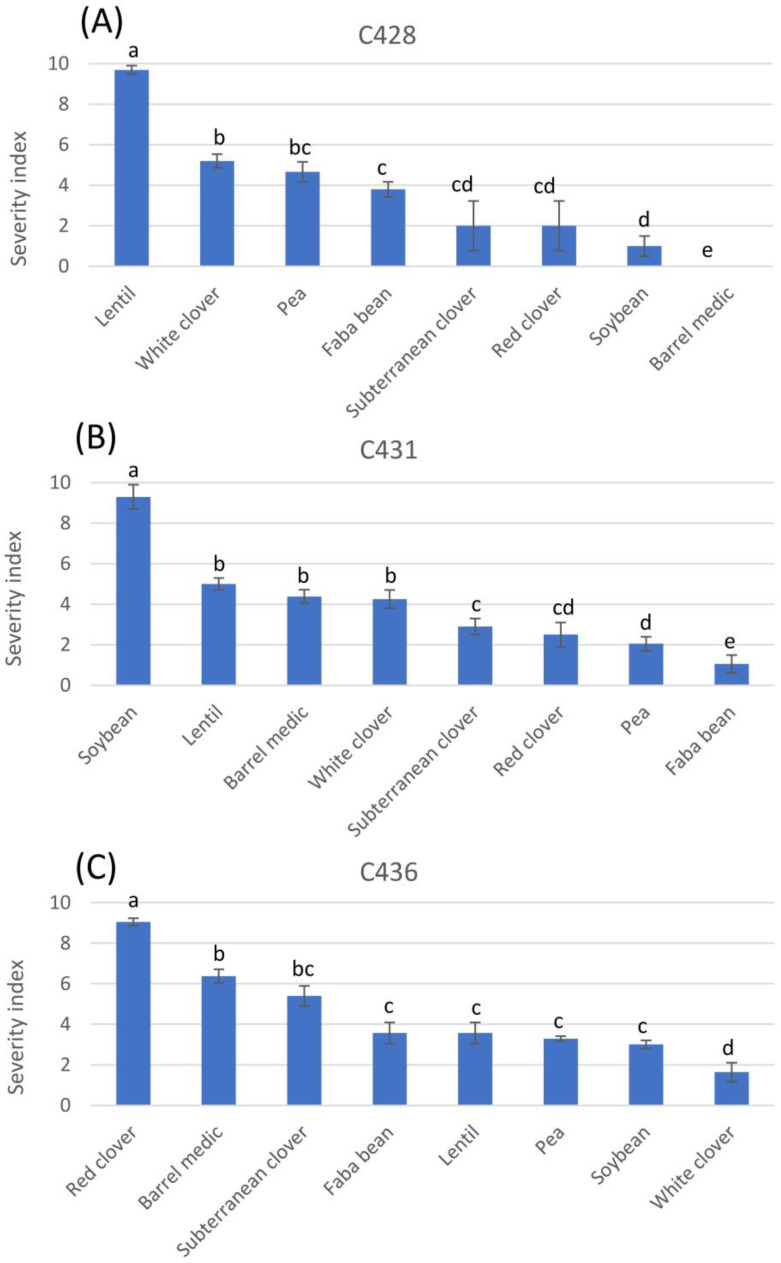
Severity index of anthracnose symptoms on legume species inoculated with (**A**) *Colletotrychum truncatum* isolate C428, (**B**) *C. truncatum* isolate C431, and (**C**) *C. trifolii* isolate C436, performed under controlled conditions, calculated following Gossen [12]. For each fungal isolate, data with the same letter, per column, are not significantly different (LSD test, *p* < 0.01).

**Figure 2 jof-09-00610-f002:**
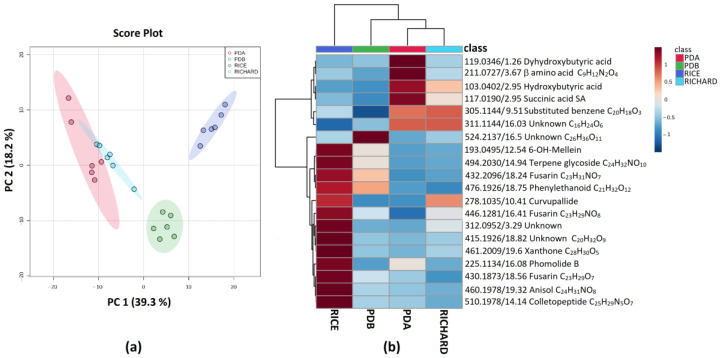
(**a**) PCA Scores plot between PC1 and PC2. The explained variances are shown in brackets; (**b**) heatmap of top 20 (ANOVA < 0.05) most different metabolites in the organic extracts of *C. truncatum* C428. The color key is based on the average peak intensity of each feature by class: red color for higher peak intensity (upregulated) and blue color for lower peak intensity (downregulated).

**Figure 3 jof-09-00610-f003:**
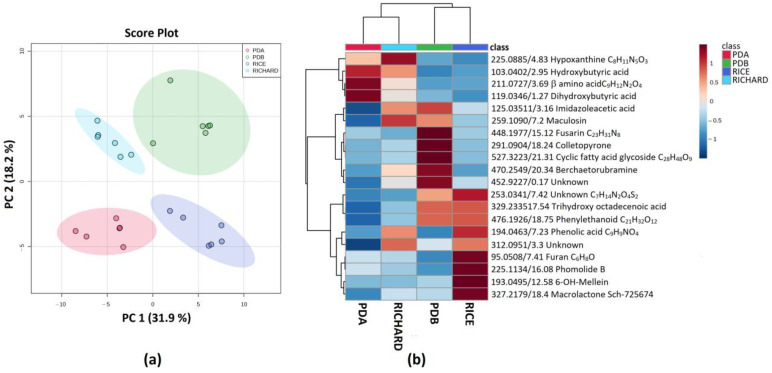
(**a**) PCA Scores plot between PC1 and PC2. The explained variances are shown in brackets; (**b**) heatmap of top 20 (ANOVA < 0.05) most different metabolites in the organic extracts of *C. truncatum* C431. The color key is based on the average peak intensity of each feature by class: red color for higher peak intensity (upregulated) and blue color for lower peak intensity (downregulated).

**Figure 4 jof-09-00610-f004:**
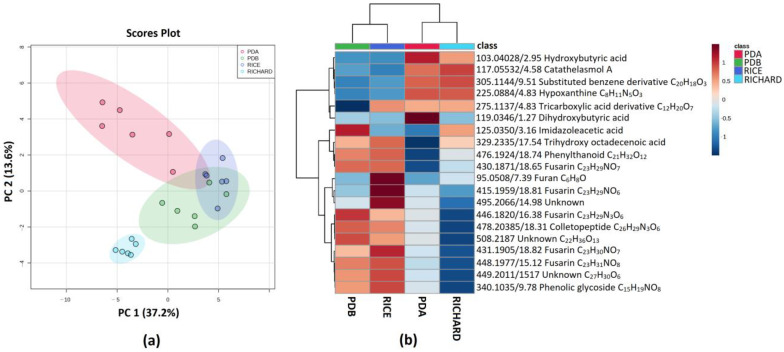
(**a**) PCA Scores plot between PC1 and PC2. The explained variances are shown in brackets; (**b**) heatmap of top 20 (ANOVA < 0.05) most different metabolites in the organic extracts of *C. trifolii* C436. The color key is based on the average peak intensity of each feature by class: red color for higher peak intensity (upregulated) and blue color for lower peak intensity (downregulated).

**Figure 5 jof-09-00610-f005:**
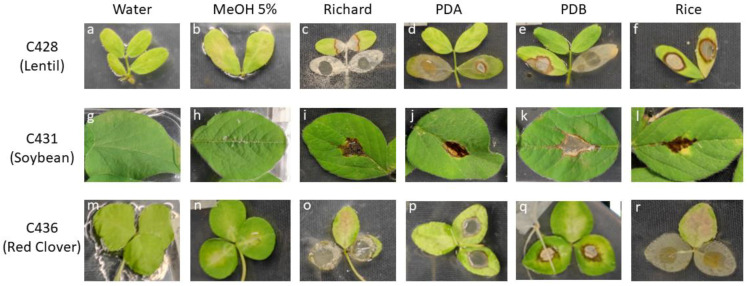
Symptoms observed on host legume leaves by *Colletotrichum* spp. culture media extracts after 13 days. (**a**,**g**,**m**) Negative control water; (**b**,**h**,**n**) negative control MeOH 5%; (**c**) effect of Richard’s medium extract of C428; (**d**) effect PDA extract of C428; (**e**) effect of PDB extract of C428; (**f**) effect of rice culture extract of C428; (**i**) effect of Richard’s medium extract of C431; (**j**) effect PDA extract of C431; (**k**) effect of PDB extract of C431; (**l**) effect of rice culture extract of C431; (**o**) effect of Richard’s medium extract of C436; (**p**) effect PDA extract of C436; (**q**) effect of PDB extract of C436; (**r**) effect of rice culture extract of C428.

**Figure 6 jof-09-00610-f006:**
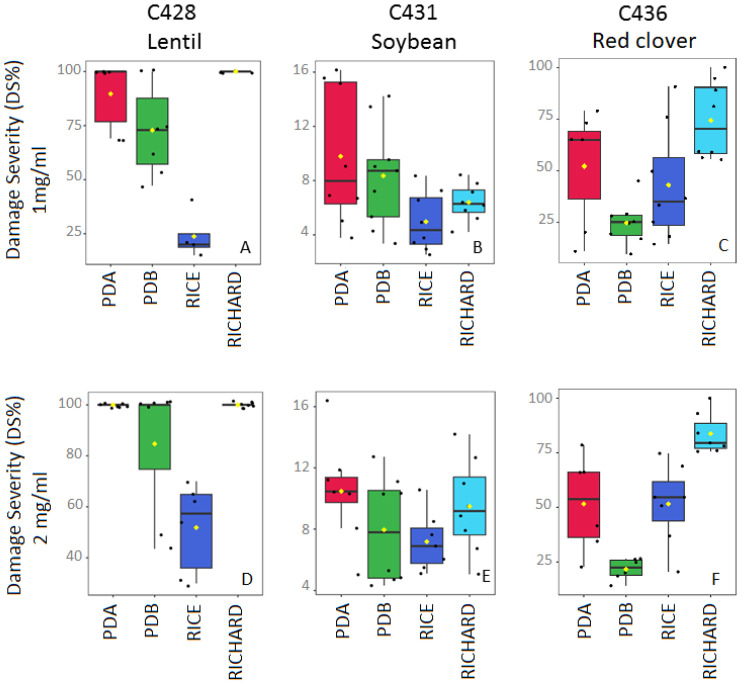
Box plots reporting the DS% of organic extracts of C428, C431, and C436 on the corresponding host, tested at 1 mg/mL and 2 mg/mL. (**A**) DS% of C428 extracts tested at 1 mg/mL on lentil; (**B**) DS% of C431 extracts tested at 1 mg/mL on soybean; (**C**) DS% of C436 extracts tested at 1 mg/mL on white clover; (**D**) DS% of C428 extracts tested at 2 mg/mL on lentil; (**E**) DS% of C431 extracts tested at 2 mg/mL on soybean; (**F**) DS% of C436 extracts tested at 2 mg/mL on white clover.

**Figure 7 jof-09-00610-f007:**
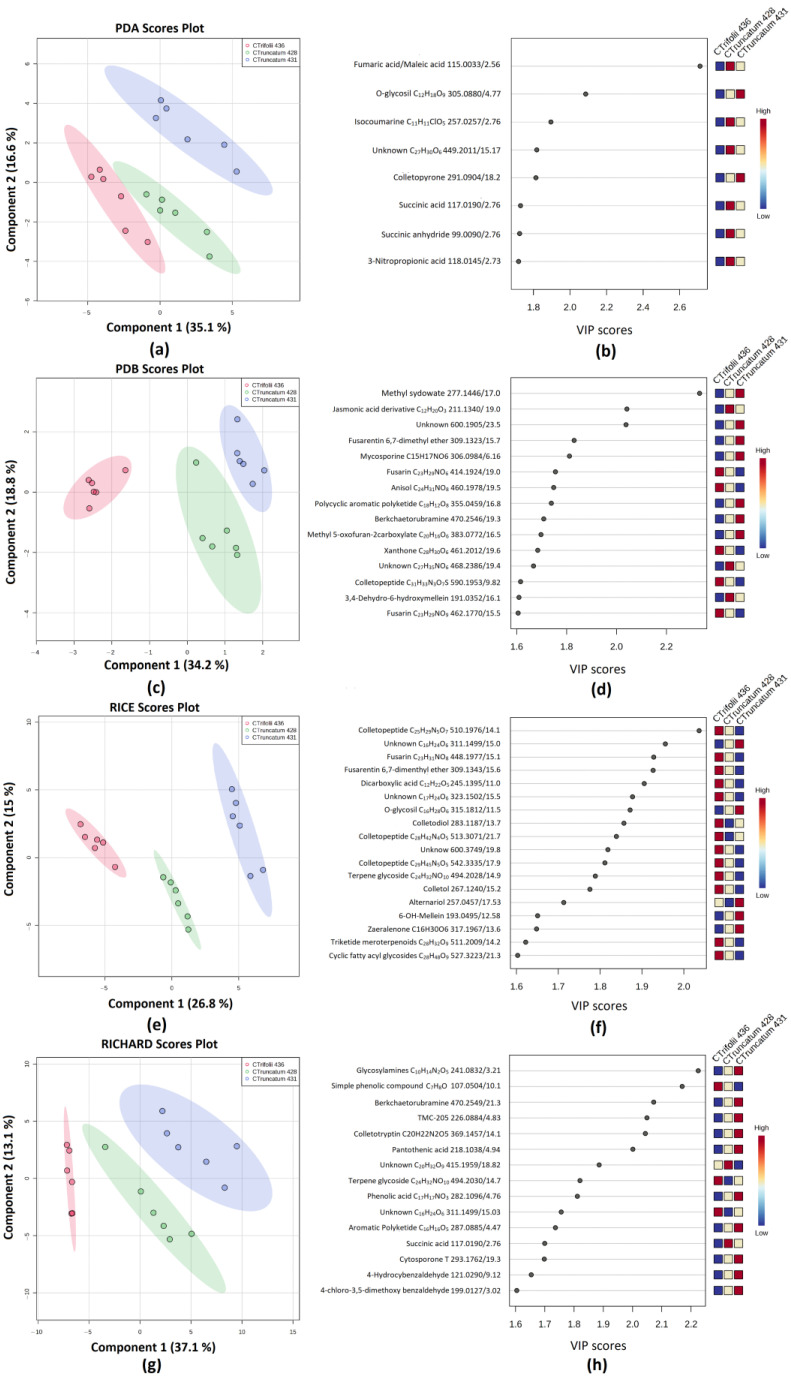
(**a**) PLS-DA scores plot between PC1 and PC2 of PDA medium extract. The explained variances are shown in brackets. (**b**) Important features identified by PLS-DA. The colored boxes on the right indicate the relative concentrations of the corresponding metabolite in *C. trifoli* C436, *C. truncatum* C428 and C431 extracts, respectively. (**c**) PLS-DA scores plot between PC1 and PC2 of PDB medium extract. The explained variances are shown in brackets. (**d**) Important features identified by PLS-DA. The colored boxes on the right indicate the relative concentrations of the corresponding metabolite in *C. trifoli* C436, *C. truncatum* C428 and C431 extracts, respectively. (**e**) PLS-DA scores plot between PC1 and PC2 of RICE medium extract. The explained variances are shown in brackets. (**f**) Important features identified by PLS-DA. The colored boxes on the right indicate the relative concentrations of the corresponding metabolite in *C. trifoli* C436, *C. truncatum* C428, and *C. truncatum* C431 extracts, respectively. (**g**) PLS-DA scores plot between PC1 and PC2 of Richard’s medium extract. The explained variances are shown in brackets. (**h**) Important features identified by PLS-DA. The colored boxes on the right indicate the relative concentrations of the corresponding metabolite in *C. trifoli* C436, *C. truncatum* C428, and *C. truncatum* C431 extracts, respectively.

**Table 1 jof-09-00610-t001:** Fungal strains from *Colletotrichum* spp. Growth in different culture media and used for the plant bioassays.

Species	Host Plant	Fungal Code
*C. truncatum*	Lentil (*Lens culinaris*)	C428
*C. truncatum*	Soybean (*Glycine max*)	C431
*C. trifolii*	Red clover (*Trifolium pretense*)	C436

**Table 2 jof-09-00610-t002:** Legume species and genotypes growth under controlled conditions and used in detached leaf assays with *Colletotrichum* spp. extracts.

Legume	Plant Specie	Genotype
Soybean	*Glycine max*	Creator
Faba bean	*Vicia faba*	Baraca
Lentil	*Lens culinaris*	Pardina
Pea	*Pisum sativum*	Messire
Barrel medic	*Medicago truncatula*	Paraggio
Red clover	*Trifolium pratense*	B1401
Subterranean clover	*Trifolium subterraneum*	E08
White clover	*Trifolium repens*	Anteria

**Table 3 jof-09-00610-t003:** Phytotoxic metabolites identified and quantified in the extracts of the selected *Colletotrichum* species.

Compound	*C. truncatum* C428				*C. truncatum* C431				*C. trifolii* C436			
	PDA	PDB	RICE	RICHARD	PDA	PDB	RICE	RICHARD	PDA	PDB	RICE	RICHARD
Colletochlorin E	n.d.	n.d.	n.d.	0.01 ± 0.004	n.d.	n.d.	n.d.	++	n.d.	n.d.	n.d.	n.d.
Colletopyrone	0.01 ± 0.001	8.13 ± 2.78	0.94 ± 0.11	+	0.04 ± 0.003	14.62 ± 1.32	0.04 ± 0.01	+	+	n.d.	0.18 ± 0.02	+
4-Hydroxybenzaldehyde	0.81 ± 0.66	0.05 ± 0.01	0.09 ± 0.02	0.59 ± 0.04	0.19 ± 0.05	0.06 ± 0.02	0.11 ± 0.02	0.74 ± 0.29	0.15 ± 0.03	0.09 ± 0.006	0.19 ± 0.03	0.49 ± 0.06
Resorcine	0.04 ± 0.02	n.d.	++	n.d.	n.d.	n.d.	n.d.	n.d.	++	n.d.	n.d.	++
Orcinol	n.d.	++	n.d.	n.d.	0.02 ± 0.002	+	n.d.	n.d.	n.d.	+	++	n.d.
Higginsianin A	n.d.	n.d.	n.d.	n.d.	n.d.	n.d.	n.d.	n.d.	+	n.d.	n.d.	n.d.
Higginsianin B	0.01 ± 0.002	++	++	++	++	+	++	++	++	++	++	+
4-Hydroxyphenyl acetic acid	0.68 ± 0.14	+	0.06 ± 0.02	0.18 ± 0.03	0.09 ± 0.05	+	0.21 ± 0.02	0.06 ± 0.01	0.79 ± 0.60	n.d.	0.03 ± 0.002	0.29 ± 0.02
6-Hydroxymellein	+	0.06 ± 0.004	0.30 ± 0.01	++	++	0.06 ± 0.004	0.41 ± 0.02	0.03 ± 0.003	+	+	++	++

n.d. = not detected; + = <LOD; ++ = <LOQ; a = concentration reported as μg/mL ± SD.

## Data Availability

Data that arose from this research are contained in the manuscript and Appendix A. Data sets and raw mass spectrometry data are available on request from the authors.

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
