# Peer review of "Uncovering Phytotoxic Compounds Produced by Colletotrichum spp. Involved in Legume Diseases Using an OSMAC–Metabolomics Approach"

_jof, 2023, doi:10.3390/jof9060610_

Round 1

Reviewer 1 Report

The manuscript describes a study focused on Colletotrichum fungi, which cause anthracnose disease in various crops and result in significant economic losses worldwide. The researchers used a method called One Strain Many Compounds (OSMAC), combined with targeted and non-targeted metabolomics profiling, to investigate the secondary phytotoxic metabolite panels produced by pathogenic isolates of C. truncatum and C. trifolii. They also assessed the phytotoxicity of the fungal crude extracts on their primary hosts and related legumes and correlated the results with the metabolite profile that arose from the different cultural conditions. The study aimed to shed light on the biologically active and structurally unusual metabolites synthesized by Colletotrichum spp. during the host infection process.

1.     There are small mistakes such as oC in Page 1 and 4. 

2.     Did the data represent in Figure 6 and Table 4 the same data set? In my opinion, authors can choose only Figure 6 and Table 4 will be moved to Supplementary. 

3.     Please check again on the reference that you cited, Frantzen et al., 1982, as I could not find the Richard medium in that reference.  

4.     “p” in p < 0.01 should be italicized when written in a manuscript or report as it is a statistical symbol.

5.     Colletotrichum trifolii in Table 1 can be shorten into C. trifolii.

6.     Colletotrichum in Sub-title 3.3 and 3.5 should be in italics format. 

Some typoes were observed. 

Author Response

Reviewer 1:

The manuscript describes a study focused on Colletotrichum fungi, which cause anthracnose disease in various crops and result in significant economic losses worldwide. The researchers used a method called One Strain Many Compounds (OSMAC), combined with targeted and non-targeted metabolomics profiling, to investigate the secondary phytotoxic metabolite panels produced by pathogenic isolates of C. truncatum and C. trifolii. They also assessed the phytotoxicity of the fungal crude extracts on their primary hosts and related legumes and correlated the results with the metabolite profile that arose from the different cultural conditions. The study aimed to shed light on the biologically active and structurally unusual metabolites synthesized by Colletotrichum spp. during the host infection process.

  1. There are small mistakes such as oC in Page 1 and 4. 

Authors: oC has been corrected along the manuscript

  1. Did the data represent in Figure 6 and Table 4 the same data set? In my opinion, authors can choose only Figure 6 and Table 4 will be moved to Supplementary. 

Authors: Following the suggestion of Reviewer 1, Table 4 has been moved to Supplementary, and the manuscript modified accordingly.

  1. Please check again on the reference that you cited, Frantzen et al., 1982, as I could not find the Richard medium in that reference.  

Authors: we thank the reviewer, we have changed the reference Frantzen et al. 1982, with:

Khan, A. I., Bhandari, R. R., Pokhrel, A., & Yadav, R. N. (2018). A study on root exudation pattern and effect of plant growth promoting fungi during biotic and abiotic stress in pigeon pea. World J Agric Res, 6(4), 122-131. DOI: 10.12691/wjar-6-4-2.

  1. “p” in p < 0.01 should be italicized when written in a manuscript or report as it is a statistical symbol.

Authors: Ok, the manuscript has been corrected accordingly.

  1. Colletotrichum trifoliiin Table 1 can be shorten into C. trifolii.

Authors: Ok, Table 1 has been corrected as suggested.

  1. Colletotrichum in Sub-title 3.3 and 3.5 should be in italics format. 

Authors: Usually, as a common editing rule, when a title is in italics, common Latin names reported in italics along the text are written in a standard format. Considering that the sub-title sections in MDPI journal format are in italics, we wrote “Colletotrichum” in the standard format.

Comments on the Quality of English Language

Some typoes were observed. 

Authors: The manuscript has been carefully revised for typos and other grammar errors.

Reviewer 2 Report

The title is clear and concise, effectively conveying the focus of the study.

The use of the OSMAC metabolomics approach is intriguing and adds novelty to the research. It would be helpful to briefly explain the OSMAC (One Strain Many Compounds) concept in the introduction to provide context for readers who may not be familiar with it.

It would be helpful to discuss the broader implications of the findings. How do the identified phytotoxic compounds contribute to our understanding of legume diseases and their management? Discuss potential applications in terms of disease control strategies or the development of novel bioactive compounds.

Overall, this study holds promise in uncovering phytotoxic compounds produced by Colletotrichum spp. involved in legume diseases using the OSMAC metabolomics approach. 

Regarding the quality of the language, the whole text needs a more careful passage to fix grammatical errors in many cases.

Author Response

Comments and Suggestions for Authors

The title is clear and concise, effectively conveying the focus of the study.

The use of the OSMAC metabolomics approach is intriguing and adds novelty to the research. It would be helpful to briefly explain the OSMAC (One Strain Many Compounds) concept in the introduction to provide context for readers who may not be familiar with it.

Authors: We thank reviewer 2 for the suggestion. To describe the OSMAC concept in the introduction, the following paragraph has been added:

“In recent years One Strain Many Compounds (OSMAC) approach has emerged as a powerful tool for exploring the chemical diversity of fungal secondary metabolites. This approach involves manipulating growth conditions, such as pH, temperature, and media composition, to elicit diverse secondary metabolites from a single fungal isolate, expanding its chemical repertoire”.

It would be helpful to discuss the broader implications of the findings. How do the identified phytotoxic compounds contribute to our understanding of legume diseases and their management? Discuss potential applications in terms of disease control strategies or the development of novel bioactive compounds.

Authors: Reviewer is right. We add now a paragraph into “Discussion” regarding a potential application of metabolites for diseases management.

Overall, this study holds promise in uncovering phytotoxic compounds produced by Colletotrichum spp. involved in legume diseases using the OSMAC metabolomics approach. 

Authors: We thank reviewer 2 for the kind words addressed to our study.

Comments on the Quality of English Language: Regarding the quality of the language, the whole text needs a more careful passage to fix grammatical errors in many cases.

Authors: The manuscript has been carefully revised for typos and other grammar errors.

Reviewer 3 Report

The submitted manuscript ‘Uncovering Phytotoxic Compounds Produced by Colletotrichum spp. involved in Legume Diseases Using an OSMACMetabolomics Approach’ from Reveglia and coworkers describes the metabolic characterization of three different Colletotrichum strains (C. truncatum isolate C428, C. truncatum isolate C431, C. trifolii C436), which are pathogenic on legumes.

I think the study is interesting, the experimental design sound and the manuscript well written. There are only some minor comments from my side, which should be addressed before publication:

- in some figures the typing is very small and hard to read (also using magnification in the PDF document). Please enlarge the letters so the reader can understand your data in detail (Figures 2, 3, 4, 6, 7)

- page 7, 3.1: I assume that the authors mean rather ‘…showed that each isolate was more effective on the legume species…’ than ‘inefective’ as it is written in the current version?

- In find it quite interesting that even among the two C. truncatum isolates the metabolic profiles are that different. I think the downside of this would be that metabolic profiling can not be used for the identification of new isolates from the field. Can you add a comment to that to the discussion?

Author Response

Comments and Suggestions for Authors

The submitted manuscript ‘Uncovering Phytotoxic Compounds Produced by Colletotrichum spp. involved in Legume Diseases Using an OSMACMetabolomics Approach’ from Reveglia and coworkers describes the metabolic characterization of three different Colletotrichum strains (C. truncatum isolate C428, C. truncatum isolate C431, C. trifolii C436), which are pathogenic on legumes.

I think the study is interesting, the experimental design sound and the manuscript well written.

Authors: We thank reviewer 3 for the kind words addressed to our study.

There are only some minor comments from my side, which should be addressed before publication:

- in some figures the typing is very small and hard to read (also using magnification in the PDF document). Please enlarge the letters so the reader can understand your data in detail (Figures 2, 3, 4, 6, 7)

Authors: The font of the text in Figures 2,3,4,6,7 have been modified and now the figures are more readable.

- page 7, 3.1: I assume that the authors mean rather ‘…showed that each isolate was more effective on the legume species…’ than ‘inefective’ as it is written in the current version?

Authors: authors wrote “Cross inoculations performed with different Colletotrychum spp. showed that each isolate was more infective on the legume species….”

- In find it quite interesting that even among the two C. truncatum isolates the metabolic profiles are that different. I think the downside of this would be that metabolic profiling can not be used for the identification of new isolates from the field. Can you add a comment to that to the discussion?

Authors: The following paragraph has been added in the discussion:

“These results could assist for species identification, enabling precise classification and enhancing our understanding of fungal diversity. Furthermore, could support ecological studies by unraveling interactions between fungi and their host plants. In our research, C428 was isolated from lentils, while C431 was isolated from soybeans, and the observed metabolic differences may reflect the level of specificity in host selection. However, further in-depth studies are required to identify appropriate biomarkers for Colletotrichum spp. chemotaxonomy.”